# Unreliability of Clinical Prediction Rules to Exclude without Echocardiography Infective Endocarditis in *Staphylococcus* *aureus* Bacteremia

**DOI:** 10.3390/jcm11061502

**Published:** 2022-03-09

**Authors:** Jorge Calderón-Parra, Itziar Diego-Yagüe, Beatriz Santamarina-Alcantud, Susana Mingo-Santos, Alberto Mora-Vargas, José Manuel Vázquez-Comendador, Ana Fernández-Cruz, Elena Muñez-Rubio, Andrea Gutiérrez-Villanueva, Isabel Sánchez-Romero, Antonio Ramos-Martínez

**Affiliations:** 1Infectious Diseases Unit, Service of Internal Medicine, Hospital Universitario Puerta de Hierro, 28222 Majadahonda, Spain; itziardiego@gmail.com (I.D.-Y.); albertomoravar94@gmail.com (A.M.-V.); nlbarrena@hotmail.com (J.M.V.-C.); anafcruz999@gmail.com (A.F.-C.); elmuru@gmail.com (E.M.-R.); a.gutierrezv@hotmail.com (A.G.-V.); aramos220@gmail.com (A.R.-M.); 2Investigational Institute Puerta de Hierro-Segovia de Arana (IDIPHSA), 28222 Majadahonda, Spain; 3Microbiology Service, Hospital Universitario Puerta de Hierro, 28222 Majadahonda, Spain; jorge050390@hotmail.com (B.S.-A.); msromero@salud.madrid.org (I.S.-R.); 4Cardiology Department, Hospital Universitario Puerta de Hierro, 28222 Majadahonda, Spain; susana.mingo.sm@gmail.com

**Keywords:** *Staphylococcus* *aureus*, endocarditis, clinical prediction rules, echocardiography

## Abstract

Background: It is unclear whether the use of clinical prediction rules is sufficient to rule out infective endocarditis (IE) in patients with *Staphylococcus* *aureus* bacteremia (SAB) without an echocardiogram evaluation, either transthoracic (TTE) and/or transesophageal (TEE). Our primary purpose was to test the usefulness of PREDICT, POSITIVE, and VIRSTA scores to rule out IE without echocardiography. Our secondary purpose was to evaluate whether not performing an echocardiogram evaluation is associated with higher mortality. Methods: We conducted a unicentric retrospective cohort including all patients with a first SAB episode from January 2015 to December 2020. IE was defined according to modified Duke criteria. We predefined threshold cutoff points to consider that IE was ruled out by means of the mentioned scores. To assess 30-day mortality, we used a multivariable regression model considering performing an echocardiogram as covariate. Results: Out of 404 patients, IE was diagnosed in 50 (12.4%). Prevalence of IE within patients with negative PREDICT, POSITIVE, and VIRSTA scores was: 3.6% (95% CI 0.1–6.9%), 4.9% (95% CI 2.2–7.7%), and 2.2% (95% CI 0.2–4.3%), respectively. Patients with negative VIRSTA and negative TTE had an IE prevalence of 0.9% (95% CI 0–2.8%). Performing an echocardiogram was independently associated with lower 30-day mortality (OR 0.24 95% CI 0.10–0.54, *p* = 0.001). Conclusion: PREDICT and POSITIVE scores were not sufficient to rule out IE without TEE. In patients with negative VIRSTA score, it was doubtful if IE could be discarded with a negative TTE. Not performing an echocardiogram was associated with worse outcomes, which might be related to presence of occult IE. Further studies are needed to assess the usefulness of clinical prediction rules in avoiding echocardiographic evaluation in SAB patients.

## 1. Introduction

*Staphylococcus aureus* (SA) bacteremia (SAB) is one of the most frequent causes of positive blood cultures (BC) and has a high mortality [1,2]. One of the most serious complications in patients with SAB is the development of infective endocarditis (IE), which worsens the prognosis [2,3]. IE in patients with SAB can occur in the absence of risk factors and with structurally normal heart valves [4].

The diagnosis of IE in patients with SAB is of vital importance because it modifies patient management. In comparison with what is recommended in other forms of SAB [5,6,7], both European [8] and American [9] IE guidelines recommend higher antibiotic doses (i.e., 12 gr/day of cloxacillin in IE versus 6 gr/day in other SAB) and longer courses (4–6 weeks for native valve IE and 6 or more weeks for prosthetic valve IE versus 2–4 weeks in other SAB). Furthermore, the addition of rifampicin and gentamicin in prosthetic valve IE is recommended by both European and American guidelines [8,9] and not recommended in other SAB. Finally, both sets of guidelines recommend cardiac surgery in case of persistent bacteremia and other complications. European guidelines even consider surgery for all SA prosthetic valve IE [8]. It has been demonstrated that approximately half of SA IE has surgical indication [3,10,11], the majority of whom have an urgent or early surgery indication. Implementation of these measures has a positive impact on prognosis of patients with IE, and this is only possible if the correct and early diagnosis is reached. Accordingly, some authors have estimated that not reaching a diagnosis of IE in patients with SAB conveys a 15–20% increase in mortality [12]. Therefore, systematic echocardiography has been classically recommended, with transesophageal echocardiography (TEE) being preferred to transthoracic echocardiography (TTE). This recommendation appears in the current guidelines [7] with a grade of evidence A-II.

This increased mortality should be weighed against the risks induced by routine TEE. Some studies have shown that serious complications of the performance of TEE is around 0.1% [13,14,15]. However, these studies are old and generally limited to special risk situation, such as esophageal varices [15] or thrombocytopenia [16]. The rate of serious complications in other scenarios is probably lower. Therefore, the estimated IE risk should be low to exclude the use of TEE [12,17].

Consequently, there have been several attempts to define subsets of patients with SAB at low risk for IE, who could be safely managed omitting the performance of TEE. Recently, three clinical prediction scores have been published, PREDICT [18], VIRSTA [19], and POSITIVE [20]. These scores can identify patients at low risk of IE in which TEE could be deferred. To date, the scarce validation studies available have yielded controversial results regarding their usefulness to exclude IE without TEE performance [21,22,23]. Moreover, all these studies have an important limitation: the low rate of echocardiography assessment, especially TEE, in groups classified as low-risk. It has been shown that the proportion of IE diagnosed increases with the number of patients with SAB who receive echocardiographic assessment [24,25,26]. Hence, given that in the development and validation studies of the scores, the majority of low-risk patients did not undergo TEE, it is likely that some patients had occult IE. This underdiagnosis could have influenced the score accuracy and, as stated above, the prognosis of SAB in mentioned studies.

Despite these controversial results and limitations, some authors recommend forgoing echocardiographic evaluation (even TTE) in patients with no identifiable risk factor [27,28]. Furthermore, a recent scenario-based survey showed that approximately 15–20% of experts would recommend not performing TTE in patients with negative VIRSTA score [29].

Therefore, our primary objective was to evaluate the PREDICT, VIRSTA, and POSITIVE scores, and to test their usefulness in safely ruling out IE without the need for TEE or TTE. The secondary objective was to determine, in patients without IE, the effect of withholding echocardiographic assessment on SAB mortality, in particular in those identified as low risk for IE, considering that missing an IE diagnosis could potentially cause an excess mortality.

## 2. Materials and Methods

We conducted a retrospective single-center study from January 2015 to December 2020. Our center is a third-level university hospital with 613 beds in Madrid and a target population of 550,000 inhabitants. At our setting, we attend approximately 60 IE episodes per year, more frequently causes by *Streptococcus* spp. and SA, followed by coagulase-negative staphylococci, *Enterococcus* spp., and Gram-negative bacilli [30].

Through a microbiology database, we identified all patients older than 18 years with a first (index) BC positive for SA during the study period. Patients with previous episodes of SAB were excluded. Index and follow-up BCs were obtained at discretion of the attending physician. BC were processed using the BD BACTEC FX system (Becton Dickinson, Sparks, MD, USA). When BC were positive, the strain was identified by MALDI-TOF (Bruker Daltonic^TM^). All systems were applied according to the manufacturer’s instructions.

Data were collected retrospectively from the electronic medical record, including demographic, comorbidities, clinical, microbiological, echocardiographic, and outcome data.

The study was approved by the hospital ethics committee. Since this was a retrospective, noninterventional study and only required collection of previously generated and anonymized data, informed consent was not required.

### 2.1. Definitions

IE was defined according to modified Duke criteria for definite IE [8]. Bacteremia was considered persistent when follow-up BC were positive at least 48 h after the extraction of the index BC despite appropriate antibiotic therapy, accordingly to the scores development studies definition [18,19,20]. Complicated bacteremia was defined according to current guidelines [7]. Primary bacteremia was defined as that in which the original source could not be determined. Sepsis and septic shock were defined according to current guidelines [31]. Relapse was defined as the appearance of a new SAB at least 15 days after negative follow-up BC or 30 days after the extraction of the index BCs and initial clinical resolution in the absence of follow-up BCs. PREDICT, POSITIVE, and VIRSTA evaluated items and cutoff point are defined previously [18,19,20] and summarized in Appendix A.

### 2.2. Predefined Ruling Out IE Thresholds

IE was considered ruled out when the risk of IE was low enough for the patient not to benefit from echocardiographic assessment. In defining cutoff points for this low IE risk, we considered recent publications that estimate the risk of IE beyond which a patient does benefit from such assessment [12,17]. We also considered the usefulness of the negative likelihood ratio (NLR) to predict this risk, as previously reported [32]. In accordance with the aforementioned studies, we predefined the following cutoff points: 1—If the risk of IE was less than 1% and the NLR was less than 0.05, IE would be considered ruled out without the need for any echocardiographic assessment; 2—If the risk of IE was between 1–2% and the NLR was less than 0.10, IE would be considered ruled out with the use of TTE without TEE. 3—If the risk of IE was between 2–5% and the NLR was less than 0.20, it would be considered uncertain if IE may possibly be ruled out with a negative TTE without performing TEE. 4—If the risk of IE was greater than 5% or the NLR was greater than 0.20, TEE would be considered necessary to rule out IE. Additionally, we considered that if the prevalence of endocarditis in low-risk patients with negative TTE was higher than 1.0%, these patients would benefit from TEE, in accordance with other authors [8].

### 2.3. Primary and Secondary Objectives

Our primary purpose was to assess the prevalence of IE in patients identified as low risk by means of PREDICT, VIRSTA, and POSITIVE score, as well as determine the NLR of these scores. Our secondary objective was to evaluate the association between not performing an echocardiogram and 30-day mortality.

### 2.4. Statistical Analysis

Quantitative variables are presented as median and interquartile range (IQR) and qualitative variables are presented as percentages and absolute values.

For the primary objective, the PREDICT, VIRSTA, and POSITIVE scores were validated by calculating their sensitivity, specificity, likelihood ratios, predictive values, and their area under the receiving operating curve (AUC). The percentage of patients identified as low-risk by these scores and who finally had IE is provided, including its 95% confidence interval (CI). Sensitivity analyses were performed including different populations according to whether different echocardiogram modalities were performed. A score was considered valid to rule out IE, with or without ETT, when the percentage of IE in patients identified as low-risk and the score’s NLR were below the mentioned cutoff points.

For the secondary objective, patients with IE diagnosed were excluded. In order to mitigate survivor bias, we excluded those patients who died within 48 h of index BC extraction. Univariate analysis of 30-day mortality was performed using chi-square for qualitative variables (or Fisher exact test) and Mann–Whitney U for quantitative variables. Multivariate logistic regression models were developed, including echocardiography assessment (TTE, TEE, or either) and those clinically relevant variables identified as statistically significant in the univariate analysis. Bilateral *p*-values of less than 0.05 were considered statistically significant. All statistical analyses were performed using SPSS version 25 statistical software (SPSS Inc., IBM, Chicago, IL, USA).

## 3. Results

During the study period, 404 patients with first episodes of SAB were identified. TTE was performed in 62.3% (250) and TEE in 32.2% (128). Fifty (50) patients (12.4%) met modified Duke criteria for definite IE. Fourteen (14) fulfilled pathological Duke criteria. All 50 patients fulfilled clinical Duke criteria for definite IE. All IE cases fulfilled the microbiological major criteria (isolation of SA in two or more blood cultures), and 46 (92.0%) fulfilled cardiac image major criteria; 33 of them with vegetation visualized in TEE, 9 with vegetation visualized in TTE and TEE not performed, and 4 of them with a positive positron emission tomography–computed tomography (3 with negative TEE and 1 with no TEE performed). Of the 4 patients who did not meet the cardiac image criteria, 2 had pathologic definite IE after cardiac surgery and the other 2 met clinical Duke criteria because they had 3 minor criteria and did not undergo cardiac surgery. None of the non-IE case fulfilled the cardiac image criteria. Accordingly, sensitivity and specificity of fulfilling the two major Duke criteria were 92.0% and 100%, respectively, with an AUC of 0.971 (95% CI 0.94–1.00) (Figure 1).

Baseline characteristics, clinical, microbiological, imaging, and outcome are summarized in Table 1.

### 3.1. PREDICT Score (Day 5 Model) Evaluation

Using the PREDICT score, 33.7% of patients (137/404) were identified as low risk. Of these, 3.6% had IE (Table 2), with a 95% CI exceeding 5% (0.1–6.9%). The validation of the score is shown in Table 3 and Figure 1. PREDICT score had a NLR of 0.27. According to the predefined cutoff points, employing this score would not be sufficient to safely rule out IE without TEE.

Table 4 shows sensitivity analyses with the percentage of IE in patients with negative PREDICT score in different population according to echocardiogram evaluation. Of the low-risk patients with a negative TTE, 3.4% (3/89) were eventually diagnosed with IE. No alternative cutoff point substantially improved the accuracy of PREDICT score (Appendix A).

### 3.2. POSITIVE Score Evaluation

Using the POSITIVE score, 60.3% of patients (245/404) were identified as low-risk. Of these, 4.9% had IE (Table 2), with a 95% CI exceeding 5% (2.2–7.7%). The validation of the score is shown in Table 3 and Figure 1. POSITIVE score had a NLR of 0.37. According to the predefined cutoff points, using the POSITIVE score would not be sufficient to rule out IE without a TEE.

Table 4 shows sensitivity analyses with the percentage of IE in patients with negative POSITIVE score in different population according to echocardiogram evaluation. Of the low-risk patients with a negative TTE, 2.8% (4/141) were diagnosed with IE. No alternative cutoff point substantially improved the accuracy of the POSITIVE score (Appendix A).

### 3.3. VIRSTA Score Evaluation

Using the VIRSTA score, 45.6% of patients (185/404) were identified as low-risk. Of these, 2.2% had IE (Table 2), with a 95% CI not exceeding 5% (2.2–4.3%). The validation of the score is shown in Table 3 and Figure 1. VIRSTA score had a NLR of 0.16. According to the predefined cutoff points, it would be uncertain if using the VIRSTA score in conjunction with a negative TTE would allow IE to be safely ruled out without a TEE.

Table 4 shows sensitivity analyses with the percentage of IE in patients with negative VIRSTA score in different population according to echocardiogram evaluation. Of the low-risk patients with a negative TTE, 0.9% (1/103) were diagnosed with IE. No alternative cutoff point substantially improved the accuracy of VIRSA score (Appendix A).

Appendix A shows the patient-level data for SAB episodes identified as low-risk by the VIRSTA score and finally diagnosed as IE. Only one patient had an identifiable risk factor (community-acquired bacteremia).

### 3.4. Echocardiographic Assessment and 30-Day Mortality

After excluding patients diagnosed with IE and those who died within 48 h after index BC extraction, performing an echocardiographic assessment was independently associated with lower 30-day mortality in a multivariate logistic regression model, including adjustment for VIRSTA score (OR 0.24 95% CI 0.10–0.54). The model is shown in Table 5. Univariate analysis for the selection of variables included is shown in Appendix A. 

Thirty-day mortality in patients classified as low-risk by VIRSTA score was lower if they had undergone echocardiographic evaluation (5.1% vs. 15.7%, *p* = 0.031).

## 4. Discussion

In our study, we evaluated whether the application of different scores for predicting IE risk in patients with SAB, with or without TTE, could safely rule out IE without TEE. Our main conclusion is that neither the PREDICT nor the POSITIVE scores would be sufficient to rule out IE without TEE, even with negative TTE. It was uncertain whether a negative VIRSTA score together with a negative TTE could allow to rule out IE without the need for TEE.

The PREDICT score was the first to be published [18]. Its high negative predictive value was ratified in a validation study performed by the same group [33]. However, other authors have found that this score does not identify low-risk patients with sufficient accuracy [20,22,23]. These results are in line with ours. It should be noted that PREDICT score does not include among its variables an important set of risk factors for IE (such as prosthetic heart valve), which could reduce its sensitivity [34]. In summary, the application of the PREDICT score to avoid performance of TEE in SAB should not be recommended.

The POSITIVE score [20] is largely based on the shorter time to positivity of BC in the case of IE. In our study, as well as in other previous work [35], despite being a risk factor for IE, the association of time to positivity with IE was not as strong as that presented by the POSITIVE score authors. In line with our findings, a recent study [23] also failed to validate this score. In summary, this score does not identify patients at low-risk with enough precision to avoid performing a TEE, so that it should not be recommended for this purpose.

The VIRSTA score [19] incorporates the largest number of variables as compared to the previous score. Consequently, some studies have shown a higher sensitivity [22,23]. Hence, a recent survey showed that 15–20% of experts would recommend not performing TTE, nor TEE, evaluation in patients identified as low-risk with this score. Yet, in our data, the VIRSTA score was not sufficient to rule out IE without performing an echocardiogram evaluation (incidence of IE in low-risk patients 2.2%, with NLR 0.16), and it was doubtful if a negative VIRSTA score plus a good-quality negative TTE would be able to rule out IE without performing a TEE. It should be noted that most of the patients eventually diagnosed with IE that had been identified as low-risk by the VIRSTA score did not have any identifiable risk factors that would have raised IE suspicion. Ruling out IE in patients with negative VIRSTA and a negative TTE would allow to avoid TEE in approximately half of patients with SAB [19,22,23]. However, well-designed studies are needed to confirm the usefulness of this approach.

A potential restriction when applying these scores is that both their development and validation studies have been performed in cohorts without universal echocardiographic assessment, and only a small proportion of patients have undergone TEE. Several authors have demonstrated that carrying out an echocardiogram in a larger proportion of patients implies a higher rate of IE diagnoses [24,25,26,36,37]. It has even been proposed that performing an echocardiogram in these patients may be associated with lower mortality [38]. In our study, we observed an increased 30-day mortality among patients with SAB who did not undergo an echocardiogram. We even found a higher mortality among patients identified as low-risk for IE who had not undergo echocardiographic evaluation. The fact that the association persists after adjusting for multiple variables, including comorbidity and severity of the bacteremia, supports the likelihood that this worse outcome was caused by the presence of undiagnosed occult IE, at least partially. Therefore, we decidedly believe that prospective studies validating these scores in a SAB cohort with systematic TTE and TEE is mandatory before assuming that they allow to avoid unnecessary echocardiography in low-risk patients.

Our study has some limitations. First, as a single-center retrospective study, it presents the limitations of external validity that this entails. However, our population is comparable to that described in recent cohorts [39]. Second, our predefined cutoff points are not prospectively validated and are based on scarce evidence. Nevertheless, both the IE risk percentages selected and the use of NLR have previously been proposed and accepted by several authors [12,17,21,29,32,37]. Third, we cannot exclude that the higher 30-day mortality found in patients with no echocardiogram is caused to uncontrolled confounding variables. However, the fact that this association persist after adjusting for several factors increases the likelihood that this worse outcome is ascribable, at least partially, to an undiagnosed IE. Finally, as mentioned, not all patients underwent echocardiography, and the percentage of patients lacking an echocardiogram was higher in low-risk patients. Of note, the majority of patients without IE did not undergo TTE or TEE. This absence of an echocardiogram, common in other studies as well, prevents the exclusion of occult IE in some of these patients, a fact that may limit the accuracy of the scores. Nevertheless, our paper adds valuable information to existing literature and provides a base for future research.

## 5. Conclusions

According to our data, none of the evaluated clinical prediction scores can replace echocardiographic evaluation, as part of the major Duke criteria, for ruling out IE. Neither the PREDICT nor the POSITIVE score were sufficient to exclude IE without concurrent TEE. In patients with negative VIRSTA score, it was uncertain whether IE could be discarded only with a good-quality negative TTE. Moreover, we found an association between not performing echocardiographic assessment and an increased 30-day mortality, even in patients at low risk for IE, which could be related to the presence of occult IE. Hence, prospective well-designed studies with systematic performance of TTE and TEE are needed to verify whether these tests can be safely avoided in any subgroup of patients.

## Figures and Tables

**Figure 1 jcm-11-01502-f001:**
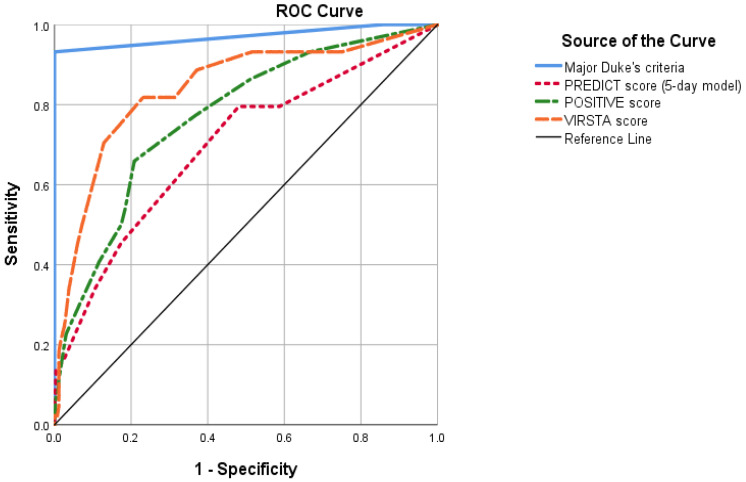
Receiving operator curve of different clinical prediction rules. AUC for PREDICT score (5-day model) was 0.699 (IC 95% 0.609–0.788). AUC for POSITIVE score was 0.771 (95% CI 0.696–0.846). AUC for VIRSTA score was 0.842 (0.771–0.912). In comparison, fulfilling major Duke’s criteria had an AUC of 0.971 (95% CI 0.94–1.00). AUC: Area under the receiving operator curve.

**Table 1 jcm-11-01502-t001:** Factors associated with IE in patients with SAB.

Variable		Total (*n* = 404)	IE (*n* = 50)	Non-IE (*n* = 354)	*p*	Missing
Demographic and comorbidity						
Age		69 (56–79)	68 (58–77)	69 (55–80)	0.927	0
Sex (female)		31.4% (127)	26.0% (13)	32.2% (1149)	0.420	0
Charlson index		2 (1–5)	2 (1–4)	2 (1–5)	0.635	3
Age-adjusted Charlson index		5 (3–7)	5 (3–7)	5 (3–7)	0.968	3
Arterial hypertension		55.9% (226)	72.0% (36)	53.7% (190)	0.015	0
Diabetes mellitus		29.2% (118)	34.0% (17)	28.5% (101)	0.506	0
Chronic heart failure		30.4% (123)	48.0% (24)	28.0% (99)	0.005	0
Ischemic heart disease		18.1% (73)	22.0% (11)	17.5% (62)	0.556	0
Natural cardiac valve disease		18.3% (74)	29.7% (22)	14.7% (52)	<0.001	0
Prosthetic heart valve disease		4.7% (19)	14.0% (7)	3.4% (12)	0.005	0
CIED		6.2% (25)	18.0% (9)	4.5% (16)	0.001	0
Chronic renal failure		22.8% (92)	28.0% (14)	22.0% (78)	0.369	0
Hemodialysis		7.1% (29)	8.0% (4)	7.1% (25)	0.794	0
Liver cirrhosis		3.2% (13)	2.0% (1)	3.4% (12)	0.715	1
Solid organ malignancy		21.6% (87)	12.0% (6)	22.9% (81)	0.098	1
Parenteral drug user		1.0% (4)	2.0% (1)	0.8% (3)	0.413	2
Clinical presentation						
Acquisition	Nosocomial	51.5% (208)	40.0% (20)	53.1% (188)	Ref.	0
	Healthcare-associated	16.8% (68)	14.0% (7)	17.2% (61)	0.870	
	Community	31.7% (128)	46.0% (23)	29.7% (105)	0.026	
Source of infection	Primary/unknown	29.4% (116)	49.0% (24)	26.7% (92)	<0.001	10
	Catheter-related	34.2% (135)	28.0% (14)	35.2% (121)	0.571	
	Other	36.3% (143)	24.0% (12)	38.1% (131)	ref	
Fever		89.8% (362)	92.0% (46)	89.5% (316)	0.635	1
Sepsis/septic shock		28.3% (114)	48.0% (24)	25.5% (90)	0.001	1
Fever defervescence within 72 h		89.2% (330)	79.2% (38)	90.6% (292)	0.050	34
Septic emboli		13.4% (54)	46.0% (23)	8.8% (31)	<0.001	0
Acute kidney injury		40.5% (162)	60.0% (30)	37.7% (132)	0.003	0
Acute cardiac failure		20.9% (84)	54.0% (27)	16.2% (57)	<0.001	2
Pitt’s bacteremia score		0 (0–3)	1 (0–3)	0 (0–2)	0.004	2
SOFA		2 (0–4)	3 (1–5)	2 (0–4)	0.038	3
Microbiology						
Time to positivity (hours)		12 (9–16)	11 (8–14)	12 (10–16)	0.023	0
Persistent bacteriemia		31.8% (99)	62.2% (28)	26.7% (71)	<0.001	93
Methicillin-resistant SAB		19.6% (79)	14.0% (7)	20.3% (72)	0.345	0
Diagnostic workup						
TTE		62.3% (250)	80.0% (40)	59.8% (210)	0.007	3
TEE		32.2% (128)	72.0% (36)	26.4% (92)	<0.001	6
PET-CT		10.2% (41)	26.0% (13)	8.0% (28)	<0.001	3
Outcomes						
30-day mortality		15.4% (62)	20.0% (10)	14.8% (52)	0.401	2
In-hospital mortality		20.3% (82)	28.0% (14)	19.3% (68)	0.187	0
SAB relapse		4.5% (17)	4.5% (2)	4.5% (15)	1.000	62

IE: Infective endocarditis. Qualitative variables are presented as percentages (absolute number) and analyzed by means of chi-square test (or Fisher exact test when necessary). Quantitative variables are presented as median (interquartile range) and analyze by means of Mann–Whitney’s *U*. SAB: *Staphylococcus aureus* bacteremia. CIED: Cardiac implantable electronic device. SOFA: Sepsis-related Organ Failure Assessment. TTE: Transthoracic echocardiography. TEE: Transesophageal cardiography. PET-CT: Positron emission tomography–computed tomography.

**Table 2 jcm-11-01502-t002:** IE and non-IE case distribution according to clinical prediction scores classification.

		PREDICT (5-Day Model)	POSITIVE	VIRSTA
High risk patients	IE	16.7% (45)	23.6% (38)	20.8% (46)
	Non IE	83.3% (224)	76.4% (123)	79.2% (181)
	Total	66.3% (269)	39.7% (161)	54.4% (221)
Low risk patients	IE	3.6% (5)	4.9% (12)	2.2% (4)
	Non IE	96.4% (132)	95.1% (233)	97.8% (181)
	Total	33.7% (137)	60.3% (245)	45.6% (185)

IE: Infective endocarditis.

**Table 3 jcm-11-01502-t003:** Validation of different clinical prediction rules to identify IE among SAB patients.

	Cut-Off	Sens.	Spec.	PPV	NPV	PLR	NLR	AUC
PREDICT (5-day model)	>1 point	90%	37.1%	16.7%	96.4%	1.43	0.27	0.70
POSITIVE	>4 points	76%	65.5%	23.6%	95.1%	2.17	0.37	0.78
VIRSTA	>2 points	92.0%	50.8%	20.8%	97.8%	1.84	0.16	0.85

IE: infective endocarditis. SAB: *Staphylococcus aureus* bacteriemia. Sens: sensitivity. Spec: Specificity. PPV: Positive predictive value. NPV: Negative predictive value. PLR: Positive likelihood ratio. NLR: Negative likelihood ratio. AUC: Area under the curve.

**Table 4 jcm-11-01502-t004:** IE rate among patients identified as low-risk by different scores in different population according to echocardiographic evaluation and results.

	PREDICT Low Risk		POSITIVE Low Risk		VIRSTA Low Risk	
	IE Prevalence	CI 95%	IE Prevalence	CI 95%	IE Prevalence	CI 95%
All patients (*n* = 404)	3.6% (5/137)	0.1–6.9%	4.9% (12/245)	2.2–7.7%	2.2% (4/185)	0.1–4.3%
Patients with TTE and/or TEE (*n* = 289)	4.8% (5/104)	0.1–9.0%	7.5% (12/160)	3.4–11.6%	3.5% (4/116)	0–6.8%
Patients with TEE (*n* = 128)	5.3% (2/39)	0–12.4%	12.0% (6/50)	2.7–21.3%	5.7% (2/35)	0–13.8%
Patients with negative TTE (*n* = 235)	3.4% (3/89)	0–7.2%	2.8% (4/141)	0.1–5.6%	0.9% (1/103)	0–2.8%

IE: Infective endocarditis. TEE: Transesophageal echocardiography. TTE: Transthoracic echocardiography.

**Table 5 jcm-11-01502-t005:** Multivariate logistic regression model of mortality and echocardiographic evaluation.

30-Day Mortality		OR	95% CI	*p*
Age (each year)		1.04	1.01–1.07	0.008
Charlson index (each point)		1.07	0.95–1.23	0.329
Unknown source of infection		3.70	1.67–8.20	0.001
SOFA (each point)		1.22	1.03–1.46	0.026
Complicated bacteriemia		2.69	1.18–6.17	0.019
Low-risk VIRSTA score		0.44	0.19–0.99	0.048
Echocardiographic evaluation	TTE and/or TEE	0.24	0.10–0.54	0.001
	TTE	0.28	0.13–0.60	0.001
	TEE	0.59	0.25–1.39	0.232

Only one echocardiographic variable at the time was included in the model. OR 95% CI and *p* values for variables other than echocardiographic evaluation are provided with the model including ETT and/or ETE. No significant different were observed within these variables when including the others echocardiographic evaluation. SOFA: Sepsis-related organ failure assessment. TTE: Transthoracic echocardiography. TEE: Transesophageal echocardiography. OR: Odds ratio. 95% CI: 95% confident interval.

## Data Availability

The data presented in this study are available on request from the corresponding author.

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
