# Peer review of "Unreliability of Clinical Prediction Rules to Exclude without Echocardiography Infective Endocarditis in *Staphylococcus* *aureus* Bacteremia"

_jcm, 2022, doi:10.3390/jcm11061502_

Round 1
Reviewer 1 Report
Dear Author:
It is a good idea indeed to verify the three Non-TEE scores PREDICT, POSITIVE, and VIRSTA for infective endocarditis (IE). As stated and very much in line with my personal view and our own research as well as in view of the current guidelines it is absolutely clear, that none of these scores should be used to rule out IE because of the devastating consequences a false negative result may have. Please be even more clear in your final statements that nothing can replace or substantially supplement the two major Duke criteria, that is: clear signs of a vegetation in a transesophageal echocardiography (in very few, very slender patients TTE may suffice) and at least two consecutive and consistent blood cultures with clear identification of a bacteria species prior to commencing antibiotic treatment. Nothing else matters. I have some suggestions: Please indicate the items of all the scores used. It is cumbersome to look into the literature, particularly for the hasty surgeon who ist not so familiar with these scores and the underlying items. It would be great if you can provide ROC-curves from the major Duke criteria also. In combination, they should accumulate to almost 100% sensitivity and specificity.
Author Response
Dear Editor and reviewers,
We thank the reviewers for their comments, that we believe will improve the quality of the article. Please, see below our point-by-point response.
It is a good idea indeed to verify the three Non-TEE scores PREDICT, POSITIVE, and VIRSTA for infective endocarditis (IE). As stated and very much in line with my personal view and our own research as well as in view of the current guidelines it is absolutely clear, that none of these scores should be used to rule out IE because of the devastating consequences a false negative result may have. Please be even more clear in your final statements that nothing can replace or substantially supplement the two major Duke criteria, that is: clear signs of a vegetation in a transesophageal echocardiography (in very few, very slender patients TTE may suffice) and at least two consecutive and consistent blood cultures with clear identification of a bacteria species prior to commencing antibiotic treatment. Nothing else matters.
Response: Thank you for your comment. We agree with the reviewer`s view. Nothing can replace the Duke criteria, especially major criteria. TEE should be pursued in the majority of patients with Staphylococcus aureus bacteremia, except, maybe, in a minority of selected patients with very low risk and a very good cardiac image quality on TTE, as stated in the conclusion. We have modified our final statement in order to be even more clear about the importance of this point.
I have some suggestions: Please indicate the items of all the scores used. It is cumbersome to look into the literature, particularly for the hasty surgeon who ist not so familiar with these scores and the underlying items.
Response: Thank you for the suggestion. We have added the items of all scores as part of the supplementary material (table S1) in order to facilitate understanding of the paper without a cumbersome literature review.
It would be great if you can provide ROC-curves from the major Duke criteria also. In combination, they should accumulate to almost 100% sensitivity and specificity.
Response: We thank the reviewer for the suggestion. After the reviewer comment we have evaluated the performance of the major Duke criteria in our series. For the cardiac image major modified Duke criteria, we included TTE, TEE, PET-CT and cardiac CT images, according to 2015 ESC guidelines (Habib et al, 2015 ESC Guidelines for the management of infective endocarditis: The Task Force for the Management of Infective Endocarditis of the European Society of Cardiology (ESC), https://doi.org/10.1093/eurheartj/ehv319).
All IE cases fulfilled the microbiological criteria (identification of Staphylococcus aureus in 2 blood cultures). Forty-six (46) IE cases fulfilled the major criteria, 33 of them with a vegetation on TEE, 9 with vegetation of TTE and TEE not performed, and 4 of them with positive PET-CT (3 with negative TEE and 1 with TEE not performed, all of them prosthetic valves). 4 IE cases did not fulfil the major cardiac image criteria, all of them fulfilled clinical defined IE with microbiological major criteria plus 3 minor criteria (fever, cardiac predisposition, and vascular phenomena in all of them). Two of these cases underwent cardiac surgery and IE was pathological confirmed, the other 2 did not underwent cardiac surgery. No patient without IE fulfilled the major cardiac image criteria. 314 (88.9%) of non-IE patients fulfilled the microbiological major criteria.
Accordingly, fulfilling both major criteria had a sensitivity of 92.0%, specificity 100%, with an AUC of 0.971 (95% CI 0.94-1.00). Negative predictive value was 99.0% and negative likelihood ratio was 0.08. We have added this information to the text and the AUC has been added to figure 1.
Reviewer 2 Report
I read the original article of Calderón-Parra et al titled “The use of clinical prediction rules without echocardiographic evaluation in Staphylococcus aureus bacteraemia may not be enough to exclude infective endocarditis and be associated with increased mortality” with a great interest. In the present article, author aim to compare the usefulness of PREDICT, POSITIVE and VIRSTA scores to rule out IE without echocardiography.
There are some major limitations related to the study:
- The title should be way simpler and shorter than what it actually is
- Line 48: A brief review and comparison of both ESC and AHA guidelines on the topic is mandatory to better rappresent the importance of a correct and fast diagnosis
- Lines 49-51: What is written may be deceptive and should be extended to ensure that the reader understands its meaning. The studies on the risks of TEE are either extremely old or are limited to very specific circumstances such as esophageal varices and thrombocytopenia.
- Lines 55-62: This paragraph should be modified to make it more comprehensible.
- Lines 75-77: When explaining your center, it would be beneficial for the reader to have some understanding of the bacteria that most often cause endocarditis in your location.
- Line 103: Given that the most recent papers you referenced are four years old, is there anything more recent on the subject?
- Line 161: Why is "Table 4" in yellow?
- Line 164: Have you attempted to examine alternative cut-off points in order to increase sensitivity and NPV?
- Line 168: Figure 1 should be translated in english
- Lines 262-274: Another weakness of the paper is the substantial disparity in patient numbers between those with and without IE.
Author Response
Dear Editor and reviewers,
We thank the reviewers for their comments, that we believe will improve the quality of the article. Please, see below our point-by-point response.
I read the original article of Calderón-Parra et al titled “The use of clinical prediction rules without echocardiographic evaluation in Staphylococcus aureus bacteraemia may not be enough to exclude infective endocarditis and be associated with increased mortality” with a great interest. In the present article, author aim to compare the usefulness of PREDICT, POSITIVE and VIRSTA scores to rule out IE without echocardiography.
There are some major limitations related to the study:
- The title should be way simpler and shorter than what it actually is
Response: Thank you for the advice. The title has been shortened and simplified.
- Line 48: A brief review and comparison of both ESC and AHA guidelines on the topic is mandatory to better rappresent the importance of a correct and fast diagnosis
Response: A brief review of both ESC and AHA guidelines of management of IE in comparison of management of other forms of SAB has been added, and the importance of a correct and prompt diagnosis has been highlighted.
- Lines 49-51: What is written may be deceptive and should be extended to ensure that the reader understands its meaning. The studies on the risks of TEE are either extremely old or are limited to very specific circumstances such as esophageal varices and thrombocytopenia.
Response: Thank you for the comment. We agree that what was written could be deceptive. We have modified the paragraph to improve the understanding of the meaning.
- Lines 55-62: This paragraph should be modified to make it more comprehensible.
Response: We have modified the paragraph in order to make it more comprehensible.
- Lines 75-77: When explaining your center, it would be beneficial for the reader to have some understanding of the bacteria that most often cause endocarditis in your location.
Response: Thank you for the suggestion. We have added information about the causes and frequency of IE at our setting.
- Line 103: Given that the most recent papers you referenced are four years old, is there anything more recent on the subject?
Response: After your suggestion, we have performed a literature review in search for more recent papers calculating or estimating cut-off points of IE risk in SAB in order to forgoing TEE or TTE. There is substantial amount of paper about estimating the IE risk factors on SAB patients (most of them cited by our work). However, none of them estimate or describe the risk-benefit balance of different echocardiographic strategies in low IE risk population. To our knowledge, only the cited two papers by Heriot et al (references 12 and 17 in the new version of the manuscript) calculate of cur-off points to decide echocardiography strategy. The rest of the manuscripts only describes different populations with certain low risk of IE, without estimating the risk-benefit balance in them. We now mention the scarce of papers calculating cut-off points as a limitation. Despite this limitation, the used cut-off points are well-accepted by several authors, and we believe that our paper adds valuable information to available literature
- Line 161: Why is "Table 4" in yellow?
Response: It was an error during the manuscript edition. We have corrected it.
- Line 164: Have you attempted to examine alternative cut-off points in order to increase sensitivity and NPV?
Response: We thank the reviewer for the suggestion. After the reviewer comment, we have examined alternative cut-off points.
For PREDICT score (5-day model) the stablished cut-off point is >1, with sensitivity (S) 90% and NPV 96.4%. The other possible cut-off point (>0) had the same sensitivity but lower NPV (95.6) because of the lower specificity (E, 37.1% vs 31.0%). A lower cut-off point wouldn’t have detected any of the 5 IE cases with negative PREDICT. Prevalence of IE with this cut-off point would have been 4.6%.
For VIRSTA score, the cut-off point is > 2, with S 92%, E 50.8% and NPV 97.8%. For the other possible cut-off points: >0: S 94.0%, E 24,0%, NPV 96.5%; >1: S 94.0%, E 33.6%, NPV 97.5%. A lower cut-off punctuation (both >0 and >1) would have detected 1 out of the 4 IE cases with negative VIRSTA. Prevalence of IE with these cut-off points would have been 4.5% and 2.5%
For POSITIVE, the stablished cut-off point is > 4, with S 76,0% and NPV 95.1%. Of note, no patient had a punctuation of 1 or 2, so other possible cut-off points were >0, >3 and >4. For other possible cut-off points: >0: S 92.0%, E 35.6%, NPV 96.9%; >3: S: 84.0%, 50.6%, NPV 95.7%. A cut-off point of >3 would have identified 4 out of the 12 IE cases with negative POSITIVE, and a cut-off point of >0 would have identified 8 out of 12 IE cases. Prevalence of IE with these cut-off points would have been 3.1% and 4.3%.
Accordingly, no other cut-off point would have substantially improved the accuracy of these scores to exclude IE. We have added this information to the paper (supplementary table S2).
- Line 168: Figure 1 should be translated in English
Response: Figure 1 have been translated in English
- Lines 262-274: Another weakness of the paper is the substantial disparity in patient numbers between those with and without IE.
Response: We thank the reviewer for this comment. We believe that the reviewer refers to the disparity in the number of patients undergoing TTE and/or TEE between those with and without IE. Indeed, this is a limitation that our manuscript shares with the papers that have developed and validated the scores. We have added this to limitation section. Nevertheless, our paper adds information to current literature and provides the base for future research.
Round 2
Reviewer 2 Report
I thank the authors for accepting my suggestions which have certainly improved the readability and scientific soundness of the paper.
Author Response
We thank the reviewer again for the suggestions.